# Leaf Age Compared to Tree Age Plays a Dominant Role in Leaf $\delta^{13}$C and $\delta^{15}$N of Qinghai Spruce (*Picea crassifolia* Kom.)

**Caijuan Li** [1,2], **Bo Wang** [1,2], **Tuo Chen** [1,*], **Guobao Xu** [1], **Minghui Wu** [1,2], **Guoju Wu** [1] and **Jinxiu Wang** [1,2]

[1] State Key Laboratory of Cryospheric Science, Northwest Institute of Eco-Environment and Resources, Chinese Academy of Sciences, Lanzhou 730000, China; licaijuan@lzb.ac.cn (C.L.); wangbo900824@lzb.ac.cn (B.W.); xgb234@lzb.ac.cn (G.X.); wumh2017@lzb.ac.cn (M.W.); guojuwu@lzb.ac.cn (G.W.); nwnujinxiu_w@163.com (J.W.)

[2] University of Chinese Academy of Sciences, Beijing 100049, China

\* Correspondence: chentuo@lzb.ac.cn

**Abstract:** Leaf stable isotope compositions ($\delta^{13}$C and $\delta^{15}$N) are influenced by various abiotic and biotic factors. Qinghai spruce (*Picea crassifolia* Kom.) as one of the dominant tree species in Qilian Mountains plays a key role in the ecological stability of arid region in the northwest of China. However, our knowledge of the relative importance of multiple factors on leaf $\delta^{13}$C and $\delta^{15}$N remains incomplete. In this work, we investigated the relationships of $\delta^{13}$C and $\delta^{15}$N to leaf age, tree age and leaf nutrients to examine the patterns and controls of leaf $\delta^{13}$C and $\delta^{15}$N variation of *Picea crassifolia*. Results showed that $^{13}$C and $^{15}$N of current-year leaves were more enriched than older ones at each tree age level. There was no significant difference in leaf $\delta^{13}$C values among trees of different ages, while juvenile trees (<50 years old) were $^{15}$N depleted compared to middle-aged trees (50–100 years old) at each leaf age level except for 1-year-old leaves. Meanwhile, relative importance analysis has demonstrated that leaf age was one of the most important indicators for leaf $\delta^{13}$C and $\delta^{15}$N. Moreover, leaf N concentrations played a dominant role in the variations of $\delta^{13}$C and $\delta^{15}$N. Above all, these results provide valuable information on the eco-physiological responses of *P. crassifolia* in arid and semi-arid regions.

**Keywords:** Leaf $\delta^{13}$C; Leaf $\delta^{15}$N; Growth stage; Environmental factors; Relative importance

---

## 1. Introduction

As one of the most powerful tools for studying plant eco-physiology, stable isotope techniques provide fundamental insights into how plants interact with and respond to biotic or abiotic environmental factors, helping us to better understand the relationship between plants and their environment [1–3]. In particular, leaf carbon isotope composition ($\delta^{13}$C), which reflects the balance between leaf conductance and photosynthetic rate [4], is widely used to analyze intraspecific or interspecific differences in photosynthetic and physiological characteristics [5,6], to measure the long-term water use efficiency under different environmental conditions and to reveal significant functional changes in plant metabolism and adaptation to various environmental stresses [7–9]. In addition, the natural abundance of $^{15}$N in leaves or roots has been proposed as an important tracer to reflect the outcome of different processes affecting $\delta^{15}$N compositions, thereby providing an integrative measure of terrestrial N processes [10–12].

However, to our knowledge, multiple previous studies have been conducted on large global or regional-scale variations in plant $\delta^{13}$C or $\delta^{15}$N values, while the understanding of $\delta^{13}$C and $\delta^{15}$N

patterns on intermediate spatial or temporal scales is rather limited [11,12]. More importantly, variations of $\delta^{13}$C and $\delta^{15}$N values during different plant development and growth stages (for example, stand age class, tree age class) have been neglected. Currently, increasing attention has been paid to investigating the significant variations in stable carbon isotopes among different plant organs, such as leaves, stems, shoots, roots, or different plant species including $C_3$ and $C_4$ plants [3,13–15]. However, research about the variations in the natural abundance of $^{13}$C and $^{15}$N with leaf habit, phenological leaf traits or leaf age class on intermediate scales remains incomplete [10,13]. For example, a previous study demonstrated that leaf age was of special interest when exploring isotope fractionation, because younger leaves show different physiological properties and mechanisms of carbon and nitrogen assimilation compared to older leaves [13,16]. Likewise, significant variations in the $\delta^{15}$N values with stand age were discussed. Li et al. [10] reported that leaf $\delta^{15}$N variations at the community, plant growth form and species levels were significantly reduced with increasing stand age over shorter times and at smaller spatial scales. In addition, Vitoria et al. [17] demonstrated that there was no significant difference in leaf $\delta^{13}$C and $\delta^{15}$N values of evergreen and deciduous species within a site. In addition, despite tremendous progress over the past few decades in investigating what causes variation in plants $\delta^{13}$C and $\delta^{15}$N values, limited studies have allowed a comprehensive explanation for the relative importance of study variables on carbon and nitrogen compositions [16,17]. Therefore, it is necessary to conduct further investigation on the natural abundance of $^{13}$C and $^{15}$N during different plant growth stages over intermediate spatial or temporal scales.

Moreover, there are a variety of other abiotic and biotic factors that control leaf $\delta^{13}$C and $\delta^{15}$N values during plant development and growth [18,19]. For example, leaf $\delta^{13}$C values changed with leaf habit, morphology, genetics and irradiance [1,16], which may reflect differences in photosynthetic water use efficiency. Current work has demonstrated that plant $\delta^{13}$C is also influenced by various environmental factors such as precipitation, humidity, soil moisture, and air temperature [18,19]. Furthermore, leaf functional elements such as nitrogen (N) and phosphorus (P) also play a key role in $\delta^{13}$C values through their indirect effects on photosynthetic capacity and the synthesis of proteins, DNA and RNA [20,21]. However, our knowledge on the relative role of these parameters in leaf $\delta^{13}$C values remains incomplete. Whether and how these established relationships could hold true in specific biomes remains largely uncertain. This uncertainty is especially true for forests of different study regions that might display contrasting responses to climate [16]. Additionally, compared to plant $\delta^{13}$C, the relationship between leaf $\delta^{15}$N and those mentioned factors has received less attention. Earlier studies have focused on the spatial or seasonal variation in plant $\delta^{15}$N values along a specific factor gradient, but did not consider the relative importance of those variables in the variations of $\delta^{15}$N values [11,12]. Thus, for detailed knowledge, additional empirical studies are required to address the relative effect of biotic or abiotic factors in the variations of $\delta^{13}$C and $\delta^{15}$N values at specific biome levels.

Qinghai spruce (*Picea crassifolia* Kom.) as a common coniferous evergreen species is widely distributed at altitudes ranging from 2300 to 3300 m in the subalpine and alpine environments of Qilian Mountains in the Northern China, in which the availability of water, nutrients and temperature is crucial for determining plant performance, abundance and distribution. It exhibits a wide tolerance to different environmental conditions and has significant ecological function in northwest China. However, limited studies have been conducted on temporal and spatial variations in the stable carbon and nitrogen isotope compositions in different aged leaves and trees of *P. crassifolia*. Therefore, it is necessary to fully understand the effects of the variables mentioned above on the $\delta^{13}$C and $\delta^{15}$N values of *P. crassifolia*.

We hypothesized that the $\delta^{13}$C and $\delta^{15}$N values of *P. crassifolia* would change with leaf age and tree age to adapt to the growth stage needs. In addition, environmental variables could contribute to the growth and eco-physiology of *P. crassifolia*. The main objectives of this study were (i) to quantify the variation in *P. crassifolia* leaf $\delta^{13}$C and $\delta^{15}$N values along the leaf age and tree age gradients and provide evidence of the physiological mechanisms underlying the variations in the $\delta^{13}$C and $\delta^{15}$N

values; (ii) to investigate the relationship between leaf $\delta^{13}$C, leaf $\delta^{15}$N and leaf nutrients; and (iii) to explain the relative role of leaf physiological properties and leaf nutrients in the variations of $\delta^{13}$C and $\delta^{15}$N.

## 2. Materials and methods

### 2.1. Site Description

The research was conducted in the Shuang Longgou region (longitude 102°17′18″–102°33′42″ E; latitude 37°18′12″–37°25′18″ N) located northwest of Tianzhu County at the eastern margin of the Qilian Mountains (Figure 1). The climate is generally characterized as a semiarid continental climate with water availability being the major abiotic factor-limiting plant growth. Mean annual precipitation is less than 400 mm and mean annual air temperature in this temperate location is approximately 1.5 °C, respectively. Moreover, the dominant forest species in this study area is *P. crassifolia*, which grows naturally with no manual management, and *Juniperus przewalskii* Kom., *Betula albo-sinensis* Burk. and *Populus davidiana* Dode are the second minor contributors to the shady understory. Additionally, the shrub species are mainly dominated by *Salix cupularis*, *Rhododendron simsii* Planch., *Caragana jubata* (Pall.) Poir. and *Spiraea alpine* Pall.

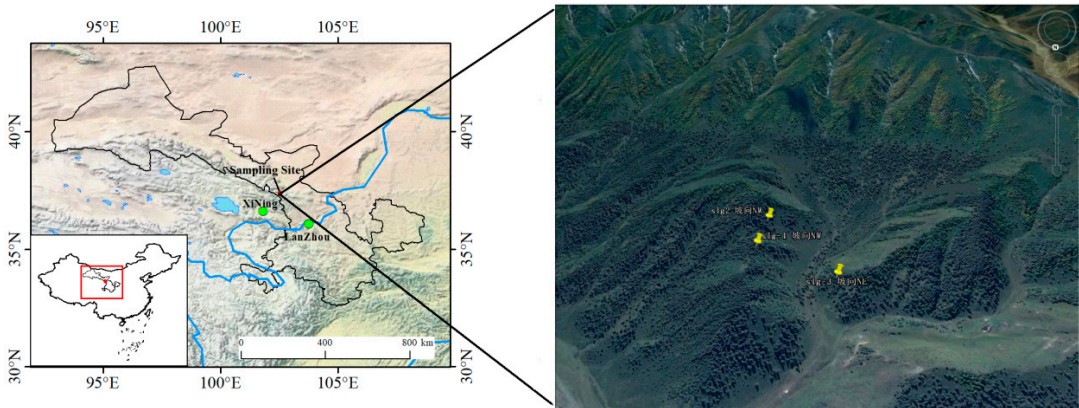

**Figure 1.** Geographic location of the collection sites (slg-2, slg-3, slg-4) of *Picea crassifolia* located in Shuanglonggou region.

### 2.2. Sample Collection

In 2015, *P. crassifolia* forests were selected from sites with similar conditions, such as topography, the composition of the undergrowth vegetation and stand age. Three plots (slg-2, slg-3, and slg-4 are shown in Figure 1) ranging from 2824 to 2914 m were established for study. Table 1 provides the general information related to the sampling sites. For each sampling plot, trees were divided into four/five groups according to the classes of diameter at breast height. Subsequently, there were five breast-height diameter classes determined and three to five trees with similar diameter in each group were chosen as the samples. The sampled *P. crassifolia* leaves developed in full light of an open canopy and were carried out from the upper third portion of the tree crown.

Moreover, considering the variation caused by differences in leaf nutrient contents at different orientations of the shoots, only leaves with a healthy appearance (avoiding damaged leaves) were cut with a pole pruner, and sampling was carried out from different orientations as much as possible. Overall, 4 years' leaves (from current year and up to 3 years-old) of each sampled tree were collected in our study. There was a clear joint on the branch between growing seasons, which aided in the determination of leaf age. We defined leaf age as 0 for the needles from the current, and then the next group was year 1 (the previous year+1) and so on. Thus, leaves aged 0–3 years were detached from the twigs before sending the samples to the laboratory for analysis. Additionally, for the estimation of tree age, tree-ring cores of selected *P. crassifolia* trees were collected at tree breast height (approximately

1.3 m above the surface) using an increment borer. Actual tree ages of sample cores were determined using dendrochronological methods [22].

**Table 1.** Site information for four sampling quadrats.

| Site code | Elevation (m) | Slope aspect | Latitude | Longitude | MH (m) | M-dbh (cm) |
|---|---|---|---|---|---|---|
| **slg-2** | 2824 | NW | 37°23′17.79″ N | 102°30′39.47″ E | 10.77 | 27.73 |
| **slg-3** | 2914 | NE | 37°23′02.92″ N | 102°30′48.26″ E | 6.53 | 17.09 |
| **slg-4** | 2828 | NW | 37°23′12.79″ N | 102°30′44.58″ E | 6.84 | 16.86 |

MH is mean tree height and M-dbh is mean diameter at breast height.

### 2.3. Stable Carbon and Nitrogen Isotope Analyses

The samples were washed in distilled water to remove dust particles, air dried before oven drying at 65 °C for 12 h and at 110 °C for 10 min to deactivate the enzymes, ground into a homogeneous fine powder, and sieved in the laboratory. A stable carbon and nitrogen analysis was performed in the Environmental Stable Isotope Laboratory, Institute of Environment and Sustainable Development in Agriculture, No.12, Zhongguancun South Street, Haidian District, Beijing 100081, China.

The isotopic compositions of the leaf samples were measured on an Isoprime100-EA mass spectrometer (Germany). The carbon or nitrogen isotope ratios are expressed relative to an international standard using the delta notation:

$$\delta_{\text{sample}} = (R_{\text{sample}} - R_{\text{standard}})/R_{\text{standard}}. \tag{1}$$

where $\delta_{\text{sample}}$ was defined by this relationship, $R_{\text{sample}}$ indicated the $^{13}\text{C}/^{12}\text{C}$ or $^{15}\text{N}/^{14}\text{N}$ ratio of the sample, and $R_{\text{standard}}$ indicated the $^{13}\text{C}/^{12}\text{C}$ or $^{15}\text{N}/^{14}\text{N}$ ratio of the standard. The international standard reference for carbon was PDB (Pee Dee Belemnite), and for nitrogen, it was an average of $^{15}\text{N}/^{14}\text{N}$ from atmospheric air [23].

### 2.4. Statistical Analysis

We analyzed the dataset by subdividing them into four groups based on leaf age (current year leaves, 1-year-old leaves, 2-year-old leaves, and 3-year-old leaves) and tree age (<50-year-old, 51 to 100-year-old, 101 to 150-year-old, and >150-year old), respectively. For leaf $\delta^{13}\text{C}$ and $\delta^{15}\text{N}$ values of different leaf ages and tree ages, the mean, median, standard error, and coefficient of variation (CV) were calculated, respectively. Here, the analysis was the leaf age–tree age combination. We first used two-way analysis of variance and Tukey's post hoc test to compare differences of leaf $\delta^{13}\text{C}$ and $\delta^{15}\text{N}$ values between leaf ages and tree ages. Next, the regression analysis was applied to investigate the relationship between leaf $\delta^{13}\text{C}$, leaf $\delta^{15}\text{N}$ and leaf nutrients (leaf N, P concentrations and the C:N ratios). Furthermore, we calculated the relative importance (refers to the quantification of an individual regressor's contribution to a multiple regression model) of each predictor on leaf $\delta^{13}\text{C}$ and $\delta^{15}\text{N}$ with the R package relaimpo [24].

The R package relaimpo demonstrates six different metrics for assessing the relative importance of regressors (all regressors are uncorrelated) in the model [24]. Each predictor's contribution is just the $R^2$ from univariate regression, and all univariate $R^2$-values add up to the full model $R^2$. $R^2$ represents the proportion of variation in y that is explained by the p regressors in the model. Correlation analysis was conducted using the SPSS 22.0 [25] and R 3.2.4 [26].

## 3. Results

### 3.1. Variations of P. crassifolia Leaf $\delta^{13}C$ and $\delta^{15}N$ Values with Leaf and Tree Ages

Changes of leaf $\delta^{13}\text{C}$ and $\delta^{15}\text{N}$ in relation to leaf age and tree age are shown in Figures 2 and 3. All leaf $\delta^{13}\text{C}$ values varied from −28.22‰ to −24.09‰, with a mean of −26.76‰ and a variance of

3.39%. The average carbon isotopic values from current year to 3-year-old leaves were −25.54‰, −27.15‰, −27.24‰ and −27.07‰, respectively (Figure 2A). The $\delta^{13}$C was significantly more enriched in current year leaves than others at each tree age level ($p < 0.01$, Figure 3A), while no differences were observed among other older leaves. Meanwhile, at each leaf age level, the $\delta^{13}$C value of mature trees (>150 years old) was lower than that of other aged trees (Figure 2A), but the difference was not significant (Figure 3A).

The mean leaf $\delta^{15}$N of all samples was −5.91‰, with a range of −8.7‰ to −2.89‰ and a variance of 16.80%. The average nitrogen isotopic values from current year to 3-year-old leaves were −4.86‰, −6.00‰, −6.15‰ and −6.55‰, respectively (Figure 2B). Leaf $\delta^{15}$N showed substantial variability among leaf ages and tree ages. Older leaves were more depleted in $^{15}$N than current year leaves at each tree age level (Figure 3B). Moreover, there was a significant difference in $\delta^{15}$N between middle-aged trees (50–100) and juvenile trees (<50) except for 1-year-old leaves ($p < 0.05$).

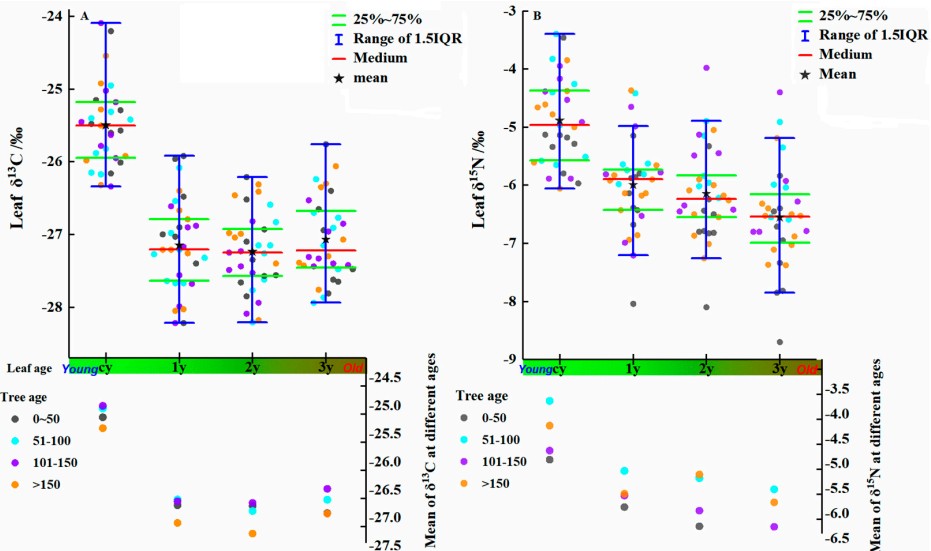

**Figure 2.** Box and scatter plots showing that the spatial patterns and variations of the $\delta^{13}$C (**A**) and $\delta^{15}$N (**B**) values changed with differently aged leaves and trees of *P. crassifolia*. Leaf ages including current year to 3-year-old leaves were represented by cy, 1y, 2y, and 3y, respectively.

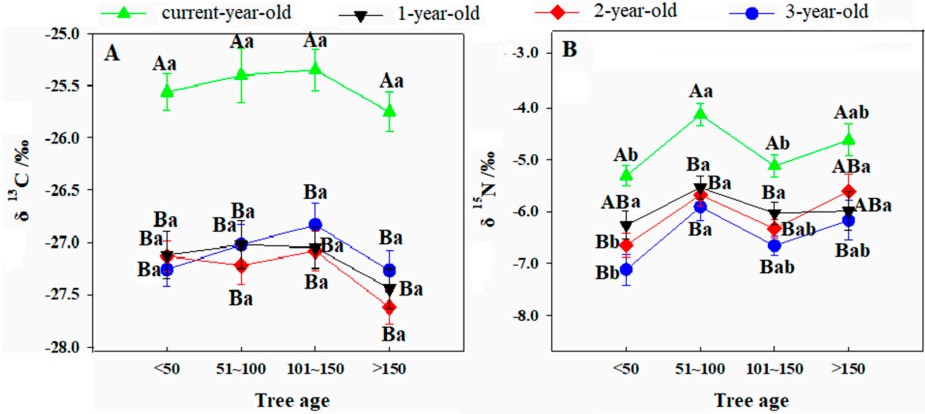

**Figure 3.** Differences in leaf $\delta^{13}$C (**A**) and $\delta^{15}$N (**B**) between 4 years' leaves collected along different tree age levels. The value was Mean± SE (standard error). Different uppercase letters represent significant differences among leaf ages at each tree age level, while different lowercase letters represent significant differences among tree ages at each leaf age level.

### 3.2. Relationship Between Leaf $\delta^{13}C$, $\delta^{15}N$ and N, P Concentrations as well as C:N Ratio

For all leaf samples, $\delta^{13}C$ was significantly positively correlated with leaf N and P concentrations, but negatively related to the C/N ratio ($p < 0.001$, Table 2). However, nutrient patterns did not differ among leaf ages as the leaf $\delta^{13}C$ showed no significant relationship between leaf N, leaf P concentrations and the C/N ratio except for current-year-old leaves ($p > 0.05$).

Likewise, leaf $\delta^{15}N$ showed significant positive correlation with leaf N and P concentrations but negative correlation with the C:N ratio in all leaf samples ($p < 0.001$). When examining the relationship between these leaf nutrients and $\delta^{15}N$ at each leaf age level, a significant positive correlation between leaf N concentration and $\delta^{15}N$ and a negative correlation between the C:N ratio and leaf $\delta^{15}N$ were only observed in current-year-old and 1-year-old leaves, respectively, while there was no significant relationship between leaf $\delta^{15}N$ and P concentrations at each leaf age group.

**Table 2.** Regression equations for leaf $\delta^{13}C$ and $\delta^{15}N$ values against leaf N and P concentrations and the C:N ratios for 4 years' leaves.

| Nutrient Variables | Leaf Ages | Statistic parameters | | | |
|---|---|---|---|---|---|
| | | $R^2(\delta^{13}C)$ | Regression coefficient | $R^2(\delta^{15}N)$ | Regression coefficient |
| Leaf N | All leaves | 0.4295 *** | 4.3035 | 0.3829 *** | 3.6156 |
| | Current-year-old | 0.0894 | −2.0361 | 0.2335 *** | 2.2635 |
| | 1-year-old | 0.0176 | −0.6792 | 0.1249 * | 1.3664 |
| | 2-year-old | 0.0007 | 0.0701 | 0.0621 | 0.4335 |
| | 3-year-old | 0.0617 | −0.5253 | 0.0006 | −0.0327 |
| Leaf P | All leaves | 0.4259 *** | 0.7824 | 0.2641 *** | 0.5462 |
| | Current-year-old | 0.0836 | −0.5255 | 0.0023 | 0.0604 |
| | 1-year-old | 0.0127 | 0.0444 | 0.0262 | 0.0477 |
| | 2-year-old | 0.0319 | 0.0491 | 0.0015 | 0.0073 |
| | 3-year-old | 0.0029 | 0.0144 | 0.0741 | -0.0473 |
| C:N | All leaves | 0.3848 *** | −19.373 | 0.3597 *** | −0.0216 |
| | Current-year-old | 0.1768 * | 5.008 | 0.2629 *** | −4.2006 |
| | 1-year-old | 0.0027 | 1.7585 | 0.2446 *** | −12.641 |
| | 2-year-old | 0.0198 | −4.9676 | 0.0597 | −5.8591 |
| | 3-year-old | 0.0131 | 3.1193 | 0.0108 | −1.8541 |

Note: *** $p < 0.001$,* $p < 0.05$.

### 3.3. Relative Importance of Leaf Age, Tree Age, Tree Height as well as Leaf Nutrients on the $\delta^{13}C$ and $\delta^{15}N$ Values

In this analysis, we assumed that the effect of leaf age may not be linear. Among the studied variables, leaf age, tree height and tree age have accounted for 18.78% variance in the model (Figure 4A). The independent effects of leaf age showed a larger contribution ($R^2 = 14.49\%$, $p < 0.05$) to total variation in leaf $\delta^{13}C$ compared with tree age and tree height. In addition, leaf nutrients such as leaf nitrogen and phosphorus concentrations were the other most important predictors for the $\delta^{13}C$ ($R^2 = 33.56\%$). In particular, leaf N concentrations explained the largest percentage of variation in leaf $\delta^{13}C$ and its effect was significant ($R^2 = 19.24\%$, $p < 0.01$).

Leaf physiological properties (leaf age, tree height and tree age) and leaf nutrients (N and P concentrations) together explained 47.29% of the variations in leaf $\delta^{15}N$ (Figure 4B). Among the study variables, leaf age was the most important predictor for the $\delta^{15}N$ ($R^2 = 15.31\%$, $p < 0.01$). Meanwhile, tree age as another physiological factor also played a key role in leaf nitrogen isotope compositions ($R^2 = 3.52\%$, $p < 0.01$). Moreover, looking at the relative contribution of each predictor in leaf nutrients, leaf nitrogen concentrations independently explained the largest percentage of variation in leaf $\delta^{15}N$ ($R^2 = 19.29\%$, $p < 0.001$).

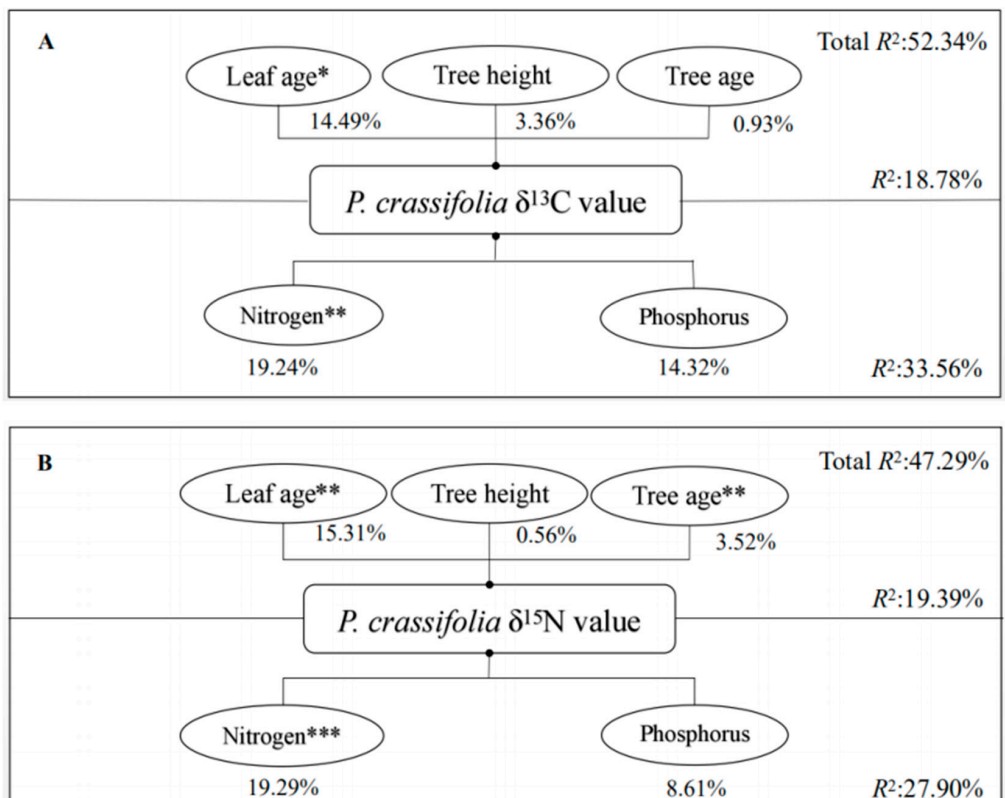

**Figure 4.** Variation partitioning (%, $R^2$) of physiological factors including leaf age, tree height and tree age; and leaf nutrients such as leaf nitrogen and phosphorus concentrations in accounting for leaf $\delta^{13}$C (**A**) and $\delta^{15}$N (**B**) values. The percentage values represent the proportion of the variance explained by each predictor in the model. *** $p < 0.001$, ** $p < 0.01$, * $p < 0.05$.

### 3.4. Relationship Between $\delta^{15}$N and $\delta^{13}$C in P. crassifolia Leaves

Relationship between the $\delta^{13}$C and $\delta^{15}$N values of all leaf ages pooled was significantly positive for *P. crassifolia* samples ($p < 0.001$, Figure 5). The correlation coefficient between them was 0.44. Different leaf ages (range from current-year leaves to 3-year-old leaves) were represented by triangles, squares, circles, and rhombuses, respectively.

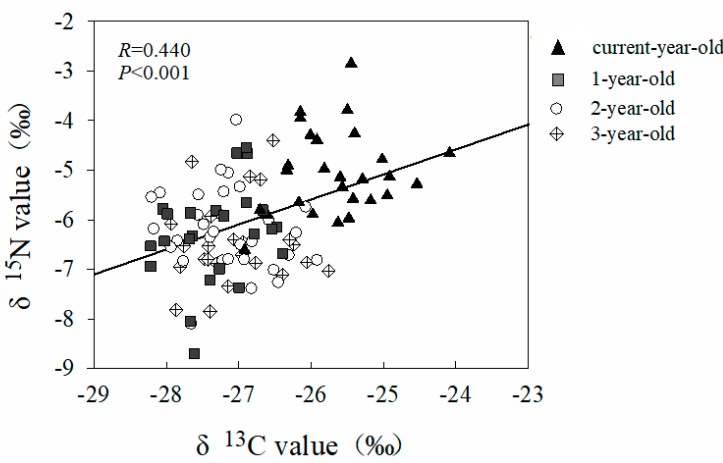

**Figure 5.** Relationship between leaf $\delta^{13}$C and $\delta^{15}$N values of *P. crassifolia*. Leaf ages (current year leaves: triangles; 1-year-old leaves: squares; 2-year-old leaves: circle; 3-year-old leaves: rhombic) are also given.

## 4. Discussion

### 4.1. Leaf δ¹³C Changed with Leaf Age and Tree Age

*P. crassifolia* is widely distributed throughout the arid zone of northwest China [22]. In our work, leaf $\delta^{13}C$ varied from −28.22‰ to −24.09‰, with a mean of −26.76‰, nearly identical to that reported by Pei et al. [27] (−28.58‰ to −25.02‰) and Yun et al. [28] (−28.9‰ to −25.4‰) for *P. crassifolia* in the Qilian Mountains. Moreover, the findings were almost the same as average values reported for *Pinus tabulaeformis* in northwest China with values of −26.82‰, which fell within the range of −28.6‰ to −25.02‰ [21].

Previous studies have confirmed that the rate of photosynthesis and respiration declines with leaf aging [13]. In this study, a significant difference in $\delta^{13}C$ between current-year leaves and other old leaves was observed. The $\delta^{13}C$ values achieved a relative maximum at the current-year leaves (−25.54‰) and lower values occurred in other old leaves. Since carbon was directly assimilated from the air or remobilized from reserve carbohydrates [13,29], we assumed that the isotopic patterns reported in our study might be caused by several effects. First of all, variations in leaf $\delta^{13}C$ values were related with *P. crassifolia* interior biochemical processes. In the initial developmental stages, a 'hungry' state of intercellular $CO_2$ concentrations exists because plants grow relatively rapidly and need to synthesize large amounts of organic matter to meet the demands of development and construction, leading to reduced distinguishing and exclusion of $^{13}CO_2$ [18]. Thus, the values become enriched. When the exterior morphology, interior structure and physiological metabolism functions are mature, plants have the ability to adjust physiological and biochemical reactions. Therefore, they can efficiently distinguish and exclude $^{13}CO_2$, and the $\delta^{13}C$ values are expected to be depleted [30]. More importantly, photosynthetic capacity is the central process that coordinates carbon isotope discrimination, with more photosynthetically active leaves being relatively $^{13}C$-enriched [16].

Second, it was associated with leaf development stage [13]. Due to the changing growth rate between different aged leaves, the allocated proportions of the structural, functional and storage components within plants varied significantly to meet the leaves' nutrient needs. Previous $^{13}CO_2$ tracer studies have reported that there were two leaf developmental stages including heterotrophic and autotrophic stages [14]. During the heterotrophic growth stage, where organic carbon was imported from elsewhere in the plant, enrichment in $^{13}C$ was most evident in this stage and supposedly a result of the heterotrophic carbon source for growth [13,31]. This stage was more obvious for current-year-old leaves. Furthermore, Cernusak et al. [32] discussed six hypotheses regarding the explanation for the $^{13}C$ enrichment of heterotrophic versus autotrophic plant organs. Based on these hypotheses, newly expanded leaves might need to synthesize large amounts of organic matter to meet the demands of development and construction, which leads to a reduction in distinguishing among sources of C and allows the leaves to obtain C from $^{13}CO_2$. Moreover, young new leaves contained more $^{13}C$-enriched cellulose and import carbon from older leaves, while the old leaves had more $^{13}C$-depleted lipids and lignin and export carbon to the younger leaves [33]. As a consequence of these effects, an enrichment of $^{13}C$ in current-year leaves was observed. During the autotrophic growth stage, carbon was assimilated and exported to other plant organs [17], which was most evident for older leaves [14]. The lighter carbon isotope was preferentially assimilated and used to produce the lipids and lignin, while the heavier carbon isotope was transported as $^{13}C$-enriched sucrose to the young new leaves [13]. As a result, the old leaves were expected to be $^{13}C$-depleted.

### 4.2. Leaf δ¹⁵N Values Changed with Leaf Age and Tree Age

Leaf nitrogen isotope compositions were determined by the isotope ratio of the external nitrogen source and physiological mechanisms within the plant. However, the intra-plant variation in isotope composition was caused by multiple assimilation events, organ-specific losses of nitrogen as well as resorption and reallocation of nitrogen [12]. In our work, $^{15}N$ of current year leaves were more enriched compared to other mature leaves—a pattern completely similar to stable carbon isotope compositions.

Meanwhile, the result of Figure 5 indicated that these ratios shift similarly to leaf age. Nitrogen as a key nutrient to build up the photosynthetic apparatus was translocated either from the roots, storage organs or mature leaves to growing leaves [13]. Therefore, there may be variation in $\delta^{15}N$ of leaves throughout the plant depending on sink/source activity and the timing and source of remobilized and assimilated organic nitrogen [34,35]. Nitrogen remobilization was important for perennial plant survival. During growth, there was significant variation in primary N-containing compounds being remobilized in the plant [34]. Masclaux-Daubresse et al. [36] reported that N-containing compounds (like proteins, chlorophyll, etc.) could be degraded during leaf senescence and then nitrogen may be remobilized from senescing leaves to expanding leaves at the vegetative stage. For example, $^{15}N$-enriched glutamine was observed as the primary transport form of organic nitrogen, which would be remobilized to developing sink leaves (receiving enriched $^{15}N$-glutamine) from a source leaf (exporting enriched $^{15}N$-glutamine) [34]. Consequently, an enrichment of $^{15}N$ in new leaves was expected. In addition, leaf proteins and in particular photosynthetic proteins of plastids were extensively degraded during senescence, providing an enormous source of nitrogen that plants could tap into to supplement the nutrition of growing organs such as new leaves and seeds [36]. Moreover, juvenile trees (<50 years old) were $^{15}N$ depleted compared to middle-aged trees (50–100 years old) at each leaf age level except for 1-year-old leaves in this study. This was likely attributed to the various allocated proportions of the structural, functional and storage components within the plant bodies to meet the plant's nutrient demands [37]. In addition, there was a significant difference in water potential, stomatal conductance, photosynthetic rate, and water-use efficiency between juvenile trees and other aged trees [37].

### 4.3. Relationships between the $\delta^{13}C$, $\delta^{15}N$ Values and Leaf Nutrients

Multiple studies have reported various correlations between the $\delta^{13}C$ and leaf nutrients [20,21,38]. In the present work, the positive relationship between $\delta^{13}C$ and N over all aged leaves together (Table 2) is in accordance with most previous studies [20,39]. Moreover, our conclusions suggest that the relative contribution of leaf N concentrations on $\delta^{13}C$ was significant ($p < 0.01$). The main cause of the positive relationships was that photosynthetic capacity increased with leaf N concentrations [20], and there was a positive correlation between leaf $\delta^{13}C$ and photosynthetic capacity [4]. However, other studies have found a negative correlation between leaf $\delta^{13}C$ and leaf N concentrations, and this was likely attributed to the presence of nitrogen-fixing species in samples such as *Caragana microphylla* [5] or an autocorrelation with water availability in a semiarid environment. In addition, in high altitude areas, low atmospheric pressure and temperature could alter the expression of the relationship between N and photosynthetic and thus, the $\delta^{13}C$-N relation [38].

Research about the relationship between leaf $\delta^{13}C$ and P concentrations is relatively limited, and the results have been inconsistent. Some studies have demonstrated the positive relationship between leaf $\delta^{13}C$ and P owing to the effect of leaf P concentrations on photosynthetic via Rubisco, while other works have found leaf $\delta^{13}C$ to be negatively related with leaf P concentration. Our study observed that leaf $\delta^{13}C$ was positively related to all aged leaves' P concentrations in simple regression (Table 2), but the effect of leaf P concentration on $\delta^{13}C$ was not significant in multiple regression (Figure 4A). This indicates that the variations in the leaf $\delta^{13}C$ values were likely caused by stomatal limitation rather than P-related changes in photosynthetic efficiency [18,20]. In addition, our findings of the negative correlation between leaf $\delta^{13}C$ and the C:N ratios is consistent with the results from multiple previous studies and suggests that *P. crassifolia* may achieve higher water use efficiency (WUE) at the expense of decreased nitrogen use efficiency (NUE) [20,38].

The uptake and discrimination of $^{15}N$ are also significantly related to plant N demand and assimilation capacity [1,40]. N availability in ecosystem, N re-translocation in plants, and N fractionation after plant uptake is known to influence leaf $\delta^{15}N$ [1,41]. Multiple previous studies have reported that there is a positive relationship between plant $\delta^{15}N$ and leaf N concentrations at various spatial scales [17]. In our work, leaf $\delta^{15}N$ was also positively related to leaf N concentrations. Furthermore, the result of Figure 4B suggested that leaf nitrogen concentrations play an importance role in accounting

for the variations of leaf $\delta^{15}$N. Changes in environmental nitrogen demand or supply could influence whole plant and organ level nitrogen isotope discrimination [12,34]. Likewise, leaf N concentrations of current-year leaves were significantly higher than other old leaves in our study area ($p < 0.001$, Figure S1). If the nitrogen supply of current-year leaves increased, discrimination could be expected to increase. However, with important questions still remaining about the relationship of leaf N and leaf $\delta^{15}$N, more comparative data are need to evaluate the potential drivers of leaf $\delta^{15}$N with increasing leaf N concentrations in the future [11].

## 5. Conclusions

In summary, the carbon and nitrogen assimilation in *P. crassifolia* leaves resulted in the same gradient of stable isotope compositions: young *P. crassifolia* leaves were more enriched in $^{13}$C and $^{15}$N compared with the older leaves at each tree age level. No significant difference in $\delta^{13}$C values among different tree ages was observed at each leaf age level, while the $\delta^{15}$N values of middle-aged (51–100 years old) were significantly more enriched than juvenile trees (<50 years old) at each leaf age level except for 1-year-old leaves. Based on the relative importance analysis, we identified that leaf age compared to tree age plays a dominant role in variation in leaf $\delta^{13}$C and $\delta^{15}$N values. Leaf nutrients such as leaf nitrogen concentrations are also important determinant factors for leaf $\delta^{13}$C and $\delta^{15}$N. However, our knowledge on the mechanism and effects of these biotic and abiotic factors on leaf $\delta^{13}$C and $\delta^{15}$N values at large scales are still limited. Further investigation is necessary to consider combinations of different drivers and their relative importance on the $\delta^{13}$C and $\delta^{15}$N values.

**Supplementary Materials:** The following are available online at http://www.mdpi.com/1999-4907/10/4/310/s1, Figure S1: Differences in leaf N concentrations (A) and leaf P concentrations (B) between four years' leaves (from current-year-old and up to 3-year-old) collected along different tree ages levels. Different uppercase letters represent significant differences among leaf ages at each tree age level, while different lowercase letters represent significant differences among tree ages at each leaf age level.

**Author Contributions:** Writing—original draft preparation, writing—review and editing, data curation, formal analysis and validation, C.L.; methodology, software, visualization and supervision, B.W.; conceptualization, project administration and funding acquisition, T.C.; investigation, resources and visualization, G.X.; investigation and software, M.W.; resources, G.W.; investigation, J.W.

**Funding:** This research was funded by National Natural Science Foundation of China, grant numbers 31670475, 41421061.

**Acknowledgments:** We appreciate three anonymous reviewers and editors for their helpful comments to improve the manuscript. We thank Gaosen Zhang for helping us with leaves sampling. We also thank American Journal Experts help us to improve the language.

**Conflicts of Interest:** The authors declare no conflict of interest.

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
