# Peer review of "Leaf Age Compared to Tree Age Plays a Dominant Role in Leaf δ13C and δ15N of Qinghai Spruce (Picea crassifolia Kom.)"

_forests, doi:10.3390/f10040310_

Reviewer 1 Report

This article examines the influence of tree age and leaf age on d13C and d15N in leaf tissue. The results are interesting and has implications for interpreting ecology-based isotope ratios. However, there needs to be some more attention to detail on understanding the possible causes for variations in isotope ratios that were observed. This is particularly important for d15N. In general, the manuscript was well written but there are sections where there are grammatical errors and the clarity is poor. I would suggest a professional editing service to improve these characteristics of the manuscript. Furthermore, there needs to be some additional detail presented in the manuscript. There are some sections that are speculative and do not use recent literature to support their statements.

While this may not be possible, it would be very helpful to have soil d15N values to get a better indication of whether there is overall d15N fractionation by the plant or by age and whether the values are converging or diverging on the soil-based d15N value.

The title is incorrect. 13C and 15N fractionation was not measured in this case. They did not provide evidence of fractionation effects at the leaf level, particularly for N where the source d15N is unknown. Here the authors measured leaf d13C and d15N.

Materials and Methods:

How were the leaves sampled from the upper canopy of tall trees?

Results and Discussion

What about the idea that fractionation can happen when carbon and nitrogen are remobilized from leaves to trunks and then back into developing new leaves. This effect can create different signatures and should be discussion.

Figure 3. Both current season d13C and d15N are enriched relative to older tissue. This could be caused by several effects. 1. This could be an effect of the current season being less mixed than previous season’s leaves which would have gone through several stages that promote remobilization of resources.

Line 229…..chemical traits in this case is only N and P concentrations. I would just be specific and state this.

Figure 4 is not clear. There should be a better way to visually display correlation coefficients for each variable. Maybe a heat map with time on the X and variable on the Y with colors used to indicate the correlation?

The nitrogen fractionation discussion is weak and doesn’t incorporate recent studies on isotope discrimination and fractionation within plants. The decrease in d15N and d13C follows theories on enriched tissue either exported to sink tissue or loss as volatilized gas at some point of the life of the leaf. Nevertheless, this is an important message for sampling of evergreen trees where multi-year processes can affect these measured values. Figure 6 indicates this where, within each leaf age group, there is no real relationship between d13C and d15N but across leaf ages, there is a relationship indicating that these ratio shift similarly as leaves age.

It would be helpful to report N and P concentrations in the manuscript rather than just its correlation with d15N.

Line 274-376: This sentence doesn’t make sense. It needs editing and a reference to support a statement like this.

Line 379….reduction in rate? What about the possibility of rooting depth affecting d15N because of changing source d15N values. Overall, this whole paragraph needs some substantial thought. It is currently too speculative and doesn’t make use of the substantial literature that discusses intraplant fractionation of nitrogen isotopes.

Author Response

We are grateful for reviewers’ patient correction of our manuscript, both scientifically and grammatically. During our revisions, we have tried to take all of your suggestions and corrections into account to improve our manuscript. We have revised some part of introduction, methods and materials, results and discussion according to your suggestions. We have addressed all your concerns and described how we revised them, but we will be happy to work with you to resolve any remaining issues. 

Reviewer 2 Report

This paper explores foliar d13C and d15N patterns from Picea crassifolia at a montane research site in northern China, and correlates them with leaf age, tree age, foliar nutrition, climate. Unfortunately, this manuscript lacks a clear question about the study species, about tree physiology, or about ecosystem processes.  Nor does the study design suggest a clear question other than how isotopes vary with leaf age and tree age.  Instead this reads more as a “fishing” exploration of a relatively limited dataset.  The other analyses, therefore, are somewhat weak, particularly the climate analysis which I believe is wholly invalid.

I’m most concerned about the climatic sensitivity analysis.  Such analyses are often carried out with tree ring time series of many decades, provided there are good climatic data to compare them with.  However, the application of such an analysis to such a short climatic and isotopic time series makes me suspicious.  It’s very weak to look for correlations with a record as short as 4 years, let alone to do this for five climatic variables across 8 months without any adjustment for multiple comparisons.  Even allowing the analysis with this caveat, the fact that many of the correlations are significant at p<0.05 is suspicious and implies that something was done incorrectly. 

The true replication is that you have for this analysis is four years of plant tissue 13C (from the four leaf cohorts), to correlate with four years of climatic data and the sampling across trees can’t be considered independent with respect to climate.  I suspect the finding of significant correlations comes from erroneously counting samples from multiple trees as independent replicates for the sake of this analysis?

Moreover, it’s probably not valid to use leaf cohorts in this way (analogous to tree rings), because there are likely physiological differences between the cohorts of different ages.  Indeed, other parts of the analysis hint at the strong effect of leaf aging.  Without sampling each cohort multiple times (in different years or in places far enough apart that they have distinct-to-independent climatic time series), there is no way to apportion the observed variation across years to climate variation vs. systematic leaf aging effects.

I’m also surprised that height is not mentioned or included as an explicit variable in the analysis … I wonder if the age pattern observed is in fact attributable to height.  See Brienen et al. (2017) and the citations within, especially those that relate to vertical profiles of foliar isotopes in a tree canopy. 

Brienen, R.J.W., Gloor, E., Clerici, S., Newton, R., Arppe, L., Boom, A., Bottrell, S., Callaghan, M., Heaton, T., Helama, S., Helle, G., Leng, M.J., Mielikäinen, K., Oinonen, M., Timonen, M., 2017. Tree height strongly affects estimates of water-use efficiency responses to climate and CO2 using isotopes. Nat. Commun. 8, 288. https://doi.org/10.1038/s41467-017-00225-z

Repeatedly throughout the manuscript, the interpretation of the patterns observed is vague and general, and offers no novel or specific insight into the physiological and biogeochemical patterns underlying the observed isotopic differences.  This is especially evident when comparing the results with those of previous studies, where “species differences” are invoked as the explanation (e.g. lines 295, 301, 389, 438)

The quality of the English is generally comprehensible, but is frequently awkward in terms of verb tense/agreement and word choice.  Should the authors submit another draft, I recommend a thorough professional editing.

Specific comments:

L13:  I don’t think it’s fair to characterize stable isotope ratios as leaf “traits”; in and of themselves, they are biologically meaningless, but are can be useful indicators of the physiological and biogeochemical processes which operate differentially on isotopes of differing masses.

L73:  again, mean temperature and mean precipitation in and of themselves are not the determinants of isotopic patterns at large scales; these are simply how we summarize the underlying operating variables of temperature and precipitation (or soil moisture) which affect physiology and C and N cycling, which in turn have detectable effects on C and N stable isotope ratios

L125:  how mature were the current-year needles at the time of sampling?  (Were they full length and thickness?  Had the cuticle formed fully?  Were they the same color as the older cohorts of needles?)  This information would be useful in interpreting the results about these leaves.  i.e. I wonder how the results might have been different if current-year needles had been sampled a bit later in the growing season.

L137:  why were cores not collected from lower on the bole, if the goal was to age each tree?  This is of little consequence though, as long as the same methods were used for all sampled trees.

 L139:  I’m not sure what “growth cones” means; maybe this is just a poor translation … It sounds like you were using an increment borer?

Table 1:  these heights seem very small for trees of such large average diameter.  Is this correct? 

L163-171:  Delta notation is the standard for stable isotope studies like this … there is no need to include so much detail on what it means, especially since the paper as currently framed .  An appropriate citation for delta notation is Coplen 2011:

Coplen, T.B., 2011. Guidelines and recommended terms for expression of stable-isotope-ratio and gas-ratio measurement results. Rapid Commun. Mass Spectrom. 25, 2538–2560. https://doi.org/10.1002/rcm.5129

L230:  here and throughout, the word “content” is used where I think we’re probably actually talking about “concentration”  … these are important to keep distinct.  A content would be in units of mg/leaf or mg/ cm2 … a concentration would be in units of mg/g.

Figure 5:  I’m concerned that not all of the predictor variables are statistically (or even definitionally) independent.  For example, in 5B, the model includes nitrogen, phosphorus, and N:P ratio.  But can you include the ratio as an independent predictor in this case?  I could make a similar case for including both N and C:N ratio; since foliar %C is very close to constant in most of the data sets I’ve worked with.

Also, before even engaging in a variance partitioning like this, I would want to see summary statistics on how much N and P concentrations varied in the sampled leaves, both across leaf age and tree age categories.  This sort of variance partitioning is only useful if the variation in the driving variables can be thought of as meaningful, rather than as noise.

L297-299:  Are these other cited studies also for P. crassifoila?  It’s not fully clear in context.

L353-355:  are there also isotopic differences between the proteins can be remobilized vs. those that remain in the senescing leaf?

L362:  when do the leaves of this species senesce?  Do they typically retranslocate N first?

Author Response

(The authors gave the same response as above.)

Reviewer 3 Report

The review of manuscript titled "Leaf age compared to tree age plays a dominant role in 13C and 15N isotopic fractionation of Qinghai spruce (Picea crassifolia Kom.)" by Caijuan Li, Bo Wang, Tuo Chen, Guobao Xu, Minghui Wu, Guoju Wu, Jinxiu Wang 
The manuscript by Tuo Chen  (corresponding author) and others aimed at investigating the influence of leaf and tree age of Picea crassifolia, climatic variables and leaf nutrients on the leaf content variation of 13C and 15N of Picea crassifolia. The authors determined the 13C and 15N isotope ratios in relation to international standards in Picea crassifolia leaves, determined physiological and biochemical mechanisms of these variations and showed climatic conditions influence on 13C and 15N isotope ratios changes. In my opinion the manuscript is interesting and presents the application of stable isotope technique in studying plant ecophysiology and response to abiotic environmental conditions. 

The Introduction and Materials and Methods sections of manuscript are written accurately. The first section of manuscript show actual state of knowledge in the subject of stable isotope technique and its using in plant physiology research. The biological material and methods used in experiments are described very precisely. Some parts of Materials and Methods although precise are very (too) long and they could be replaced by references. In my opinion Results and Discussion sections contains some deficiencies which require attention and should be corrected.

Abstract
The abstract is well written. It contains essential results, and the most important conclusions. The aims and hypothesis of the work are well described. 
Additional keywords

I think that two initial keywords (Stable carbon isotope composition (δ13C); Stable nitrogen isotope composition (δ15N) are too long. It should be shortened.

Materials and methods

The research methods are described very exactly but in a very long part of manuscript. It should be shortened. In my opinion it contains unnecessary fragments. For example: 1) the authors described the method of the estimation of tree age very exactly (line 136-144). This method is well known and commonly used in forest research. I think it should be shortened and only most important things should be given, 2) similarly in the case of marking of envelopes with collected leaves (Line 133-135), 3) the authors described stable isotope analysis very exactly. They describe not necessary things as in the lines 166-169. 
Results
In the result section of manuscript the Figures should be modified. In Figure 2 the authors show too many statistical parameters (25-75%, 1.5 IQR, medium (it should be median?) and mean). They also presents results of all leaf δ13C and δ15N analysis. Additionally the means values are placed on the symbol representing the results of the analysis. It makes this figure unclear. In Figure 3 there is no description of the error bars. Are their standard deviation or standard error bars. There are no marking of significant differences among means for tree and leaf ages. They could be marked by different lowercase or uppercase letters. In Figure 5 the authors showed the variation partitioning (in %%) of different parameters (leaf age, tree age ….) in accounting for leaf δ13C and δ15N values. There are no marking of significant differences among these values. 
Discussion
This section of the manuscript is interesting and well written, although it seems that it is too long. The author analysed several factors which influenced the values of  δ13C and δ15N. In some places they repeated the data presented earlier in the Result section. I think it is not necessary and it should be removed. Each factor is analysed separately. It causes some information to be given several times even though it could be combined.

Author Response

We are grateful for reviewers’ patient correction of our manuscript, both scientifically and grammatically. During our revisions, we have tried to take all of your suggestions and corrections into account to improve our manuscript. We have revised some part of introduction, methods and materials, results and discussion according to your suggestions. We have addressed all your concerns and described how we revised them, but we will be happy to work with you to resolve any remaining issues.

Round  2

Reviewer 1 Report

There are still come grammatical errors that need correcting but they are relatively minor. The authors did a nice job in addressing the concerns of all three reviewers.

Author Response

We are grateful for reviewers’ patient correction of our manuscript, both scientifically and grammatically. During our revisions, we have checked and corrected some grammatical errors according to your suggestions.

Reviewer 2 Report

A number of important improvements have been made to this version of the manuscript, including methodological clarifications, data corrections to in Table 1, improved information about the studies to which the results are compared, and corrections to English grammar which improve the clarity of several parts of the manuscript.

However, despite substantial changes made to the climate analysis, I fear that it is still statistically invalid, due to the very small number of years examined, and the inappropriate representation of samples of the same year of foliage from many trees as independent observations of the effect of all the climatic variation observed in each year.  To properly do this analysis, observations of each leaf cohort’s 13C should be made in multiple successive years so that the effects of year of formation can be statistically separated from the effects of age.  Observations across a consistent set of trees should be averaged – the true unit of analysis the leaf cohort. 

At first, it might seem that the variance partitioning shown in Figure 5 is getting around this problem, by attributing some of the variation to leaf age and the rest to climate.  However, this rests on the assumption that any effect of leaf age is linear, which is likely a poor assumption, given that first-year leaves are not fully formed and are even visually quite different from the other years’ leaves.

I recommend removing the climate analyses from this manuscript entirely, as I think the results are more likely to be misleading than useful.  If the remaining factors are retained in the variance partitioning, it should be acknowledged that the effect of leaf age may not be linear as assumed. 

A qualitative discussion of the effects of interannual variation may still be appropriate in the context of the findings relative to leaf age, but the statistical support to draw conclusions about the effects of such climatic variation is very weak and this should be acknowledged. 

The remaining components of this manuscript are relatively strong and still merit publication, but I can not support the publication of this manuscript if the results shown in Fig 4 (and Fig 5 as currently presented) are retained.

Author Response

We are grateful for reviewers’ patient correction of our manuscript, both scientifically and grammatically. During our revisions, we have tried to take all of your suggestions and corrections into account to improve our manuscript. We have revised some part of results and discussion according to your suggestions. We have addressed all your concerns and described how we revised them, but we will be happy to work with you to resolve any remaining issues.

Reviewer 3 Report

The manuscript has been significantly improved. The Authors have clarified the aspects requested by providing missing information or changing the parts of the previous version of the manuscript. The Results and Discussion sections of manuscript have been enriched with fragments more accurately describing the changes in leaves of Picea crassifolia individuals differing in their age. The figures were also redrawn which greatly influenced the level of the manuscript.

Below two minor edits:

Line 19-20: the sentence is unclear, something missing here, may be it should be „… were more enriched than older ….”

Line 186: the sentence is unclear: „Current year leaves were 15N enriched in 15N compared to older leaves at each tree age level (Figure. 3B).” I think there is too many  „15N”.

Author Response

We are grateful for reviewers’ patient correction of our manuscript, both scientifically and grammatically. During our revisions, we have revised some sentences according to your suggestions.
